# Impact of Overdiagnosis on Long-Term Breast Cancer Survival

**DOI:** 10.3390/cancers11030325

**Published:** 2019-03-07

**Authors:** Jean Ching-Yuan Fann, King-Jen Chang, Chen-Yang Hsu, Amy Ming-Fang Yen, Cheng-Ping Yu, Sam Li-Sheng Chen, Wen-Hung Kuo, László Tabár, Hsiu-Hsi Chen

**Affiliations:** 1Department of Health Industry Management, College of Healthcare Management, Kainan University, Taoyuan 338, Taiwan; jeanfann@ntu.edu.tw; 2Department of Surgery, National Taiwan University Hospital, Taipei 100, Taiwan; kingjen@ntu.edu.tw (K.-J.C.); brcancer@gmail.com (W.-H.K.); 3Graduate Institute of Epidemiology and Preventive Medicine, College of Public Health, National Taiwan University, Taipei 100, Taiwan; bacilli65@gmail.com; 4School of Oral Hygiene, College of Oral Medicine, Taipei Medical University, Taipei 110, Taiwan; amyyen@tmu.edu.tw (A.M.-F.Y.); samchen@tmu.edu.tw (S.L.-S.C.); 5Department of Pathology and Graduate Institute of Pathology and Parasitology, Tri-Service General Hospital, National Defense Medical Center, Taipei 114, Taiwan; cpyupath@yahoo.com.tw; 6Graduate Institute of Life Sciences, National Defense Medical Center, Taipei 114, Taiwan; 7Department of Mammography, Falun Central Hospital, 791823 Falun, Sweden; laszlo@mammographyed.com; 8Innovation and Policy Center for Population Health and Sustainable Environment, College of Public Health, National Taiwan University, Taipei 100, Taiwan

**Keywords:** overdiagnosis, mammography screening, invasive breast cancer, zero-inflated Poisson regression model

## Abstract

Elucidating whether and how long-term survival of breast cancer is mainly due to cure after early detection and effective treatment and therapy or overdiagnosis resulting from the widespread use of mammography provides a new insight into the role mammography plays in screening, surveillance, and treatment of breast cancer. Given information on detection modes, the impact of overdiagnosis due to mammography screening on long-term breast cancer survival was quantitatively assessed by applying a zero (cured or overdiagnosis)-inflated model design and analysis to a 15-year follow-up breast cancer cohort in Dalarna, Sweden. The probability for non-progressive breast cancer (the zero part) was 56.14% including the 44.34% complete cure after early detection and initial treatment and a small 11.80% overdiagnosis resulting from mammography screening program (8.94%) and high awareness (2.86%). The 15-year adjusted cumulative survival of breast cancer was dropped from 88.25% to 74.80% after correcting for the zero-inflated part of overdiagnosis. The present findings reveal that the majority of survivors among women diagnosed with breast cancer could be attributed to the cure resulting from mammography screening and accompanying effective treatment and therapy and only a small fraction of those were due to overdiagnosis.

## 1. Introduction

While the prognosis of breast cancer (BC) has been substantially improved due to early detection of breast cancer attributed to the widespread use of mammography, the issue of overdiagnosis resulting from mammography screening has been debated over the past decade [1,2,3,4,5]. As these overdiagnosed cases are biologically indolent and non-progressive they would have never progressed to clinical phase and caused death due to breast cancer during the patients’ lifetime, implying that any treatment was unnecessary and would not have been administered had screening not been applied to these women [6,7,8,9,10].

The previous studies on the extent of overdiagnosis were estimated by excess incidence due to screening compared with background incidence derived from randomized control trials or predicted incidence extrapolated from previous unexposed epochs, making allowance for lead-time [2,9,11,12]. Note that these previous methods, while estimating the proportion of overdiagnosis, require individual normal and incident breast cancer data and also a strong assumption of lead-time distribution. These traditional approaches cannot be used for assessing the impact of overdiagnosis on long-term survival when only information on breast cancer cases and deaths from breast cancer is available.

Here, we propose a new approach to estimating overdiagnosis using information on the survival of breast cancer detected by different modalities (detection modes) together with prognostic factors with the premise that overdiagnosis of BC would not result in deaths from breast cancer. However, the survivors of these overdiagnosed BCs are often indistinguishable from those with of non-overdiagnosed BC cases but without potential of dying from breast cancer due to effective initial treatment and therapy, namely the completely cured. Both types are regarded as non-progressive BC with zero-probability of dying from BC but have manifestly different causes. To distinguish the completely cured patients from overdiagnosed ones requires information on detection mode such as screen-detected cases, interval cancers, and cancers in non-participants. The overestimation of cumulative survival due to the zero-probability of dying from breast cancer resulting from overdiagnosis would be expected if these overdiagnosed cases cannot not be separated from the completely cured.

Moreover, the non-progressive BCs indicated above would also be mixed up with progressive BC patients still alive at each specific follow-up timepoint. Whether and when these progressive cases would die from BC is highly dependent on subsequent treatments and therapies and prognostic factors [13,14,15,16,17,18]. However, only relying on these prognostics may not be sufficient to distinguish between progressive and non-progressive BC because excellent survival tumors with good prognostic factors may also be a consequence of overdiagnosis due to mammography [19].

The aim of this study is therefore to apply the zero-inflated regression model to estimate the proportion of overdiagnosis resulting from mammography screening separated from the proportion of the completely cured due to effective treatment and therapy. We also assess the cumulative survival after correction for the zero-inflated part of overdiagnosis.

## 2. Materials and Methods

### 2.1. Study Subjects and Design

We quantified the respective contributions of overdiagnosis attributable to mammography and cures due to early detection and effective treatment by using a cohort composed of 1346 patients diagnosed with invasive BC at Falun Central Hospital of Dalarna County in Sweden in two periods with available information on prognostic factors, from 1996 to 1998 and from 2006 to 2010, in combination with a zero-inflated model design and analysis. The main reason of selecting two periods is mainly due to available information on immunohistochemical (IHC) markers, particularly HER2, which had not been widely tested before 2005. The period of 1996–1998 was a pilot phase for collecting such information. The two cohorts were followed over time until the end of 2010. Note that breast cancer service screening program with mammography has been offered since 1985 at the close of the Swedish Two-county randomized controlled trial [20].

In addition to longitudinal follow-up data, the current study design illustrated in Figure 1 is based on the concept of the zero-inflated model for solving the problem of being unable to distinguish between overdiagnosed cases from cured cases due to effective treatment and therapies as mentioned in the introduction. All diagnosed breast cancers are classified into three types according to the potential for progression and the cure after initial treatment. The top left circle represents overdiagnosed cases (blue) with zero probability of dying from breast cancer mainly resulting from mammography screening. The dotted box is composed of those breast cancers with potential of progression, which are further divided into two types, the cured after initial treatment (green) and the cured after subsequent therapies during 15-year follow-up (red). The final column is the estimated attributable proportions among three types of survivors of breast cancer. If there is a lack of information on detection mode it is very difficult to distinguish the cured from the overdiagnosed. The screened cohort together with the collection of these prognostic factors provide an opportunity to distinguish overdiagnosis from the cured. The derivation of percentages among breast cancer cases delineated in Figure 1 is elaborated in the Statistical Analysis section and Appendix

### 2.2. Detection Mode Related to Curation and Overdiagnosis

There are three detection modes, screen-detected cases, interval cancers, and cancers from non-participants or outside the age ranges of screening. Here we assume overdiagnosis of BC due to mammography screening can only result from screen-detected cases as they were detected though mammography. Interval cancers after the exposure to a previous screen with negative findings were detected either through possible self-referral of patients or due to the presence of symptoms and signs. Cancers from non-participants or outside screening were diagnosed due to the presence of symptom and signs. In this sense, interval cancers would enhance awareness of being diagnosed as BC compared to cancers from non-participants. This can be supported by the fact that interval cancers have higher survival than cancers from non-participants [21]. Suppose treatment and therapies were administered to three groups according to the indication for the choice of treatment modality based on significant prognostic factors. The difference of zero probability on death from BC between screen-detected cancers and interval cancers would provide information on excess zeros due to overdiagnosis resulting from mammography. The difference of zero probability between interval cancers and cancers from non-participant offers information on overdiagnosis due to increased awareness. Details of the calculation are given in the statistical section.

### 2.3. Prognostic Factors

We collected factors responsible for progressive BCs including conventional tumor attributes (size, lymph node involvement, and histological grade), three immunohistochemical markers (ER, PR, HER2), triple negative (defined by these three IHC markers), surgical treatment and adjuvant therapy. Conventional tumor attributes have been collected since the dawn of the service screening. Surgical treatment (breast-conserving surgery, mastectomy, or others), and adjuvant therapy (radiotherapy, chemotherapy, or hormone therapy) had been collected since 1996.

Data on tumor phenotypes related to IHC markers including ER, PR and HER2 status were collected retrospectively for the period of 1996 to 1998 by standard antibody staining in the largest invasive tumor component for each patient and was described in full in previous studies [22]. The antibodies (supplier, type, dilution) used for staining are delineated as follows: ER (clone SP1; Ventana Medical Systems, Tucson, AZ, USA; 1:200 dilution), PR (clone PgR 636; Dako, Glostrup, Denmark; 1:50 dilution), and HER2 (code A 0485; Dako; 1:250 dilution). The cut-off point for ER and PR positivity is nuclear staining >10% of tumor cells. The criteria of HER2 positivity was offered by manufacturer. Triple negative BC is defined as a breast tumor with all ER, PR, and HER2 being negative.

### 2.4. Statistical Analysis

Descriptive data are presented with frequency and percentage. For categorical data, the chi-square test was used to compare the difference between groups and the Fisher Exact test was used if any count was less than 5. We first applied the Poisson regression model assuming the number of BC deaths follows a Poisson distribution. We estimated follow-up women years from the date of diagnosis as BC to the date of death from BC, loss of follow-up, or the end of this study as the offset in Poisson regression model. The value of deviance divided by degree of freedom provides an indicator to assess the extent of over-dispersion and under-dispersion for the specified Poisson regression model. For the elucidation of overdiagnosis in BC, we applied the zero-inflated Poisson (ZIP) regression model [23], which is a mixture of a Poisson regression model (count part) and a logistic regression model (zero part) as derived in Appendix A. The former model (Poisson regression model, count part) was used to evaluate the prognostic factors for progressive BCs. The prognostic factors included three conventional tumor attributes and treatment and therapies (surgery, chemotherapy, radiotherapy, hormone therapy). The latter model (logistic regression model, zero part) was used to estimate the probability of zero part (including overdiagnosis cases or cure after initial treatment and therapies) for non-progressive BCs. We used the detection mode of cancers as covariates to distinguish two types in the zero part (Appendix A). We then used the regression coefficients of logistic regression part in the ZIP model to calculate the probability of zero among all BCs and respective probabilities of zero by detection mode as detailed in Appendix A. The probability of overdiagnosis due to mammography screening and enhanced awareness is calculated as follows:The probability of overdiagnosis due to mammography screening = ((The probability of zero for screen-detected – the probability of zero for interval cancer) × The probability of zero among all BCs)(1)
The probability of overdiagnosis due to awareness = ((the probability of zero for interval cancers – the probability of zero for cancers from non-participants) × the probability of zero among all BCs)(2)
The probability of cure due to treatment = The probability of zero among all BCs – ((1) + (2))(3)

We further derive 15-year cumulative survival curves with and without correcting for overdiagnosis by using the hazard rate derived from the ZIP model and the corresponding figure from the conventional Poisson regression model without considering overdiagnosis as described in Appendix A). Two-sided *p*-value less than 0.05 was treated as statistical significance. All analyses were conducted with SAS version 9.4 (SAS Institute, Cary, NC, USA).

### 2.5. Ethics Approval

This study was approved by the Joint Institutional Review Board of Taipei Medical University (TMU-JIRB, approval numbers N201607008).

## 3. Results

Table 1 shows the frequencies of age at diagnosis, first generation prognostic factors (tumor size, node status, histologic malignancy grade), IHC markers (ER, PR, HER2, triple negative), and treatment and therapies by BC death. The distribution of age at diagnosis was similar between women who died from BC and those who did not. The distributions of tumor size, status of node involvement, histological grade, ER, PR, triple negative, surgery, and hormonal therapy were significantly different (*p*-value < 0.05) according to BC death. Women who had tumor size larger than 20 mm, positive nodes, grade 3, ER(−), PR(−), and triple negative were more likely to die from BC.

Table 2 shows that conventional tumor attributes were significant predictors in both univariate and multivariable models. The crude RR was significantly higher for tumor with size 20–29 mm (9.32; 95% CI, 3.33–26.13) and 30 mm+ (13.65; 95% CI, 4.81–38.74) compared with size 1–9 mm, tumor with node positive (3.70; 95% CI, 2.46–5.57) compared with node negative, tumor with grade 3 (2.97; 95% CI, 1.99–4.43) compared with grade 1/2, and triple negative (3.32; 95% CI, 2.11–5.24) compared with non-triple negative cancers. In the multivariable analysis, tumor with size 20–29 mm (aRR = 2.63; 95% CI, 1.38–5.02) and 30+ mm (aRR = 2.39; 95% CI, 1.19–4.80) were at greater risk than those with size 1–9 mm. Positive node led to an elevated risk (aRR = 1.86; 95% CI, 1.18–2.94) as opposed to negative node after adjusting for variables related to treatment such as surgery, chemotherapy, radiotherapy, and hormonal therapy. Interpretation of effect size on treatment and therapies should be taken with great caution as they are not a reflection of efficacy of treatment and therapies but an indication for treatment and therapies according to tumor attributes. These accounted for the findings that those with mastectomy and radiotherapy had higher hazard of dying from breast cancer and insignificant effective chemotherapies and tamoxifen therapy even after adjustment for other significant prognostic factors.

The value of deviance divided by the degree of freedom, an indicator for assessing the level of over-dispersion, was about 0.46–0.59 in the univariate model and 0.49 in the multivariable model. As this value was less than 1, it strongly suggests the problem of under-dispersion (excess zeros).

We used data with complete information (*n* = 1233) on conventional tumor attributes, variables related to surgery and adjuvant therapy, and detection mode of BCs for the ZIP model analysis. The larger the value of odds ratio (OR), the higher probability to be cured after initial treatment or overdiagnosis. The larger the value of relative risk (RR), the higher the risk of dying from BC. Table 3 shows the estimated parameters, ORs, and RRs for the ZIP model.

Tumor size, node status, grade were significant factors related to risk of dying from BC after considering treatment. Compared with non-participants and outside screening of BCs, screen detected cancers and interval cancers were with higher odds (OR = 2.38, 95% CI: 0.97–5.85 and OR = 1.23, 95% CI: 0.48–3.17, respectively) of being zero.

The probability of zero part among all non-progressive BC was 56.14%. The corresponding probabilities for screen detected cancer, interval cancer, and refuser/outside screening cancers were 66.42%, 50.50%, and 45.40% respectively, which gave 8.94% overdiagnosis due to mammography screening and 2.86% due to high awareness for those interval cancers but exposed to mammography screening based on the equation (1) and (2). The probability of zero due to the curation resulting from early detection and effective treatment was 44.34% (Figure 1, green).

The 15-year prognosis-adjusted cumulative survival of BC after correcting for overdiagnosis fell from 88.25% (Figure 2, cross mark) to 74.80% (Figure 2, hollow circle) after further adjustment for prognostic factors in the count part of progressive BC (Figure 1, red). The 15-year survival rate among 43.86% progressive BC after subsequent treatments and adjuvant therapies was 32.11% after adjustment for significant prognostic factors (Figure 1, pink).

## 4. Discussion

The long-term prognosis of BC has been substantially improved over the past three decades due to early detection, mainly through mammographic screening. However, the harm of overdiagnosis is a concomitant risk of the benefit of mammography screening and it has now become a debatable issue and concern for population-based mammography screening over the past decade [1,2,3,4,5]. For breast cancer cases with overdiagnosis, there is 0% probability of dying from BC and treatment is unnecessary for them. It may also result in the overestimation of cumulative survival attributed to effective treatment and therapies in accompany with early detection of mammography screening. The survival of BC would thus be artificially inflated if such zero-inflated overdiagnosis is included. Estimating the quantity of overdiagnosis separated from the cured due to treatment is intractable but indispensable and can be truly a reflection of early detection and effective treatment and therapy. Our novel approach with the zero-inflated design and model for separating the cured from the overdiagnosed provides a solution but the conventional statistical model could not distinguish the completed cured after initial treatment (green, Figure 1) and the curation after subsequent therapies during 15-year follow-up (red, Figure 1). From the viewpoint of methodology, the use of the zero-inflated model enables us to separate the zero part with potential of progression but completely cured after initial treatment from the non-zero part with potential of progression but cured after subsequent therapies during 15-year follow-up particularly when tumor attributes related to breast cancer progression were considered in the non-zero (progressive) part.

In addition to the assessment of the impact of overdiagnosis on long-term survival, our proposed zero-inflated model also provides an insight into the proportion of overdiagnosis resulting from mammography screening that has been well studied in previous studies using excess incidence approach with lead-time adjustment [2,9,11,12]. After reviewing the primary articles that estimated the overdiagnosis level in European population-based mammography screening programs, Puliti et al. found that the rates of overdiagnosis of invasive BCs due to mammography screening varied from 0 to 54% [11]. Morrel et al. reported lower estimated baseline incidence resulted in higher level of overdiagnosis (42% vs. 30%) [24]. They also reported that longer lead-time (5 years vs. 2.5 years) contributed to lower extent of overdiagnosis (42% vs. 51%) [24]. Different background incidence rates and the assumption of lead-time distribution may account for such a wide range of estimates on overdiagnosis reported before. Several studies reported that the overdiagnosis rate was different by age [25,26]. In addition to the disparity in the methodology of lead-time adjustment and the extent of mammography screening, variation of overdiagnosis across age may also be explained by the fact that background incidence rate and the distribution of lead-time also vary with age [2,9,11,12].

Our proposed alternative approach to evaluating the extent of overdiagnosis dispenses with background incidence of BC and the assumption of lead-time distribution. We only used empirical data on BCs with available information on detection mode, treatment and therapies, and prognostic factors collected from an organized service screening program after population-based randomized controlled trial on mammography screening since 1977 in Falun (also known as Dalarna now, and Kopparberg in the 1990s), Sweden [27]. This empirical data is well suited to estimate the overdiagnosis from mammography screening and enhanced awareness as the attendance rate of mammography screening was over 80% and women in this county were also with high awareness of being diagnosed as BC through interval cancers [4,6]. Information on BCs with various detection modes is therefore useful for separating the completed cured from overdiagnosis.

It is very interesting to note that the probability of being zero part among interval cancers was higher than refuser/outside screening BCs. The difference might result from high awareness of detecting BCs through interval cancers because they had been exposed to mammography screening. Our result showed about 3% overdiagnosis due to enhanced awareness of detecting BC through interval cancers.

There are two limitations of the current study. Although the application of ZIP enables us to estimate the attributable proportions of three types of breast cancer survivors, personalized prediction for three types cannot be achieved without more updated information on molecular and imaging biomarkers can be included in the zero part and non-zero part, respectively. The second is related to the validation of this zero-inflated model by the application of the proposed model together with the estimated parameters to independent prospective follow-up data of this cohort in the future and also to data outside this country. We therefore strongly suggest here that our proposed zero-inflated model had better be applied to other countries in Europe where mammography screening programs have been widely served since the 1990s and the screening rate was also high in order to see whether and how the cure, overdiagnosis, and the survival of progressive BCs vary with different service screening programs. We also suggest that our model can be applied to regions with lower mammography screening rates and lower awareness of detecting BC in contrast to the current data with high careening rate and enhanced awareness in order to test the generalizability of our proposed zero-inflated regression.

## 5. Conclusions

In conclusion, the zero-inflated model design is a novel approach to correcting cumulative survival of early-detected BC inflated due to the zero part of overdiagnosis. Application of this model to the Dalarna breast cancer service screening program revealed that, among all breast cancers detected from this program, there were 76% survivors (44% completely cured and 32% still alive) due to early detection of mammography and effective treatment after 15 years of follow-up and overdiagnosis accounted for 12% of survivors.

## Figures and Tables

**Figure 1 cancers-11-00325-f001:**
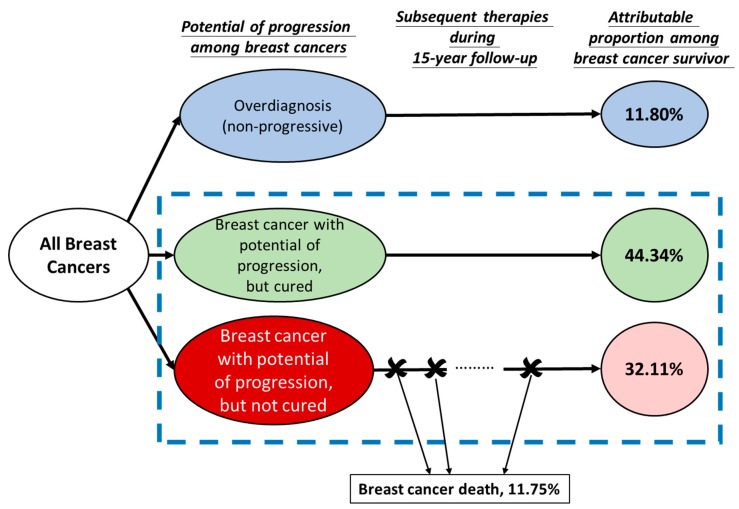
Study design for estimating the proportion of breast cancer survivors attributed to overdiagnosis, the completely cured after initial treatment, and the curation after subsequent therapies during 15-year follow-up.

**Figure 2 cancers-11-00325-f002:**
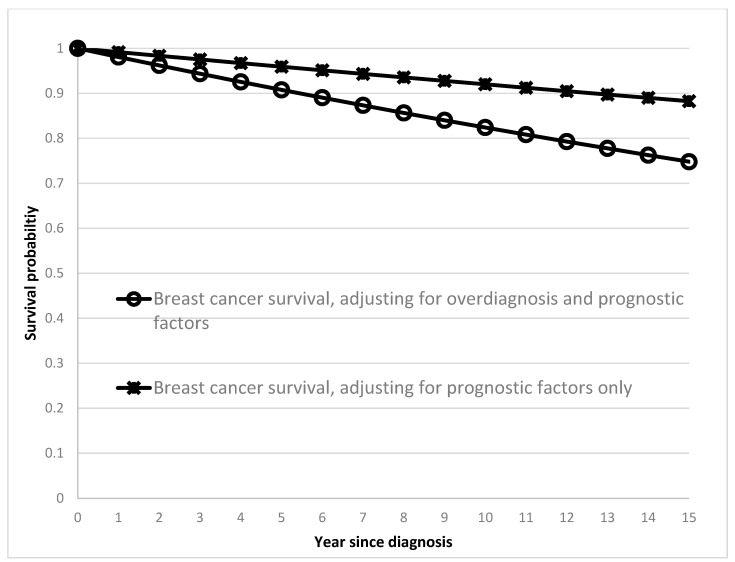
Cumulative survival of breast cancer-based models with and without considering overdiagnosis.

**Table 1 cancers-11-00325-t001:** The distribution of age at diagnosis, conventional tumor attributes, IHC markers (ER, PR, HER2, Triple negative), mammographic appearance, and treatment by status of breast cancer death.

Variable/Level	Breast Cancer Death				*p*-Value
	No (*n* = 1228)	%	Yes (*n* = 118)	%	
Age at diagnosis					0.345
<50	202	92.7	16	7.3	
50–69	596	91.8	53	8.2	
70+	430	89.8	49	10.2	
Size *, mm					<0.001
1–9	233	98.3	4	1.7	
10–14	273	96.1	11	3.9	
15–19	260	95.2	13	4.8	
20–29	263	87.4	38	12.6	
≥30	155	83.8	30	16.2	
Nodes *					<0.001
Negative	805	95.4	39	4.6	
Positive	390	87.2	57	12.8	
Grade *					<0.001
1	284	97.3	8	2.7	
2	633	93.6	43	6.4	
3	263	85.4	45	14.6	
ER *					<0.001
Negative	174	84.1	33	15.9	
Positive	990	94.6	57	5.4	
PR *					<0.001
Negative	448	87.3	65	12.7	
Positive	714	96.6	25	3.4	
HER2 *					0.8771
Negative	1018	92.9	78	7.1	
Positive	149	92.5	12	7.5	
Triple negative *					<0.0001
Yes	115	81.6	26	18.4	
No	1046	94.2	64	5.8	
Surgery *					<0.0001
MA	452	87.8	63	12.2	
BCS	538	95.9	23	4.1	
Others	238	88.1	32	11.9	
Chemotherapy					0.2018
Yes	270	89.4	32	10.6	
No	958	91.8	86	8.2	
Radiotherapy					0.8979
Yes	632	91.3	60	8.7	
No	596	91.1	58	8.9	
Tamoxifen					0.0061
Yes	480	93.9	31	6.1	
No	748	89.6	87	10.4	

Abbreviations: ER: estrogen receptor; HER2: human epidermal growth factor receptor 2; IHC markers: immunohistochemical markers; PR: progesterone receptor; BCS: breast conserving surgery; MA: mastectomy. * 66 subjects had no information on tumor size (44 survivors, 22 deaths), 55 subjects had no information on nodal involvement (33 survivors, 22 deaths), 70 subjects had no information on histological grade (48 survivors, 22 deaths), 92 subjects had no information on ER status (64 survivors, 28 deaths), 94 subjects had no information on PR status (66 survivors, 28 deaths), 89 subjects had no information on HER2 status (61 survivors, 28 deaths), 95 subjects had no information on triple negative status (67 survivors, 28 deaths).

**Table 2 cancers-11-00325-t002:** The univariate and multivariable analysis of Poisson regression model for predicting breast cancer death by conventional tumor attributes and other predictors.

Variable/Level	Univariate			Multivariable		
	cRR (95% CI)	*p*-Value	Deviance/df.	aRR (95% CI)	*p*-Value	Deviance/df.
Tumor size, mm		<0.001	0.46		<0.001	0.49
10–14 vs. 1–9	2.53 (0.80–7.93)			1.01 (0.45–2.24)		
15–19 vs. 1–9	3.12 (1.02–9.56)			1.12 (0.52–2.43)		
20–29 vs. 1–9	9.32 (3.33–26.13)			2.63 (1.38–5.02)		
30+ vs. 1–9	13.65 (4.81–38.74)			2.39 (1.19–4.80)		
Node (+) vs. (−)	3.70 (2.46–5.57)	<0.001	0.46	1.86 (1.18–2.94)	0.007	
Grade 3 vs. 1/2	2.97 (1.99–4.43)	<0.001	0.48	1.32 (0.84–2.07)	0.228	
Triple negative Yes vs. No	3.32 (2.11–5.24)	<0.001	0.47	1.53 (0.89–2.63)	0.132	
Surgery MA vs. BCS	4.02 (2.49–6.48)	<0.001	0.55	2.79 (1.56–4.98)	<0.001	
Chemotherapy Yes vs. no	1.58 (1.05–2.37)	0.027	0.59	0.83 (0.51–1.38)	0.474	
Radiotherapy Yes vs. no	0.71 (0.50–1.02)	0.063	0.59	1.39 (0.82–2.37)	0.215	
Tamoxifen Yes vs. no	0.96 (0.64–1.45)	0.849	0.59	0.89 (0.56–1.42)	0.633	

Abbreviations: aRR: adjusted relative risk; cRR: crude relative risk; df.: degree of freedom; MA: Mastectomy; BCS: Breast-conserving surgery.

**Table 3 cancers-11-00325-t003:** The regression coefficient of Zero-inflated Poisson regression model and overdiagnosis rate.

Variable	Regression Coefficient	S.E.	RR/OR (95% CI)	*p*-Value
**Count Part**	**RR**			
Intercept	−6.216	0.830		
Size, mm				0.015
10–14 vs. 1–9	1.307	0.808	3.69 (0.76–18.01)	
15–19 vs. 1–9	1.348	0.802	3.85 (0.80–18.53)	
20–29 vs. 1–9	2.329	0.769	10.26 (2.27–46.33)	
30+ vs. 1–9	2.246	0.791	9.45 (2.01–44.49)	
Node (+) vs. (−)	0.877	0.315	2.40 (1.30–4.45)	0.005
Grade 3 vs. 1/2	0.484	0.276	1.62 (0.94–2.79)	0.080
Surgery MA vs. BCS	0.651	0.360	1.92 (0.95–3.88)	0.071
Triple Negative Yes vs. No	0.914	0.311	2.49 (1.36–4.59)	0.003
Chemotherapy Yes vs. No	−0.238	0.319	0.79 (0.42–1.47)	0.456
Radiotherapy Yes vs. No	0.210	0.367	1.23 (0.60–2.53)	0.568
Tamoxifen Yes vs. No	−0.054	0.281	0.95 (0.94–1.64)	0.847
**Zero Part**	**OR**			
Intercept	−0.185	0.381		
Detection mode				0.041
SD vs. RF	0.867	0.459	2.38(0.97–5.85)	
IC vs. RF	0.205	0.484	1.23(0.48–3.17)	

Abbreviations: S.E.: Standard error; MA: Mastectomy; BCS: Breast-conserving surgery; SD: screen detected cancer; IC: interval cancer; RF: refuser & outside screening cancers.

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
