# Peer review of "Impact of Overdiagnosis on Long-Term Breast Cancer Survival"

_cancers, 2019, doi:10.3390/cancers11030325_

Round 1
Reviewer 1 Report
The first sentence in the Introduction is incorrect. Survival is the wrong term in this context, as it is a biased measure of screening efficacy, it should be mortality, though many will dispute that even with the substitution the sentence is correct. A similar error is made in the first part of the Discussion.
It is quite unclear from where the percentages in Figure 1 were derived. This should be made explicit. Further the distinction between the green and red is theoretical, even with modern biomarkers of prognosis there has to be uncertainty. This will be even more obscure for the period studied during the screening period of the Two-County trial, as Tabar revealed some time ago in a rather obscure Japanese journal, adjuvant chemotherapy was not prescribed for breast cancer in Sweden during that period.
Author Response
1 (Q) The first sentence in the Introduction is incorrect. Survival is the wrong term in this context, as it is a biased measure of screening efficacy, it should be mortality, though many will dispute that even with the substitution the sentence is correct. A similar error is made in the first part of the Discussion.
(A) The purpose of this manuscript is not to evaluate the efficacy of screening. The sentence is therefore revised as follows.
“While the prognosis of breast cancer (BC) has been substantially improved due to early detection of breast cancer attributed to the widespread use of mammography, the issue of overdiagnosis resulting from mammography screening has been debated over the past decade.” (Introduction, Page 1, Line 43-45)
“The long-term prognosis of BC has been substantially improved due to early detection mainly through mammographic screening over the past three decades.” (Discussion, Page 8, Line 244-245)
2 (Q) It is quite unclear from where the percentages in Figure 1 were derived. This should be made explicit.
(A) The Figure 1 has been re-sketched and the contexts have been also re-written as follows. Figure 1 sketches an illustration of study design taking the concept of the zero-inflated model into account.
(Please refer to point-by-point response in PDF format)
Figure 1. Study design for estimating the proportion of breast cancer survivors attributed to overdiagnosis, the completely cured after initial treatment, and the curation after subsequent therapies during 15-year follow-up
The paragraph of Figure 1 has been revised as follows.
“In addition to longitudinal follow-up data, the current study design diagrammed in Figure 1 is based on the concept of the zero-inflated model for solving the problem of being unable to distinguish overdiagnosed cases from the cured due to effective treatment and therapies as mentioned in the introduction. All diagnosed breast cancers are classified by three types according to the potential of progression and the curation after initial treatment. The top left circle represents overdiagnosed cases (blue) with zero probability of dying from breast cancer mainly resulting from mammography screening. The dotted box is composed of those breast cancers with potential of progression, which are further divided into two types, the cured after initial treatment (green) and the curation after subsequent therapies during 15-year follow-up (red). The final column is the estimated attributable proportions among three types of survivors of breast cancer.” (Study Subjects and Design, Page 2, Line 90 to Page 3, Line 99)
3 (Q) Further the distinction between the green and red is theoretical, even with modern biomarkers of prognosis there has to be uncertainty.
(A) Yes, the conventional statistical model could not distinguish the completed cured after initial treatment (green, Figure 1) and the curation after subsequent therapies during 15-year follow-up (red, Figure 1). However, the use of the zero-inflated model enables us to separate the zero part with potential of progression but completely cured after initial treatment from the non-zero part with potential of progression but cured after subsequent therapies during 15-year follow-up particularly when tumor attributes related to breast cancer progression were considered in the non-zero (progressive) part. This strength has been delineated in the Discussion (Page 8, Line 254-261)
4 (Q) This will be even more obscure for the period studied during the screening period of the Two-County trial, as Tabar revealed some time ago in a rather obscure Japanese journal, adjuvant chemotherapy was not prescribed for breast cancer in Sweden during that period.
(A) Breast cancer service screening program with mammography has been offered since 1985 at the close of the Swedish Two-county randomized controlled trial. The current analysis is based on data on invasive breast cancer cases diagnosed in the full-grown service screening program between the period of 1996 to 1998 and 2006 to 2010, both of which have already followed the standard clinical protocol of breast cancer treatment and therapy unlike the trial period. This point is not a concern.

Reviewer 2 Report
Thanks for letting me review this manuscript.
comments:
This is a good concept and a very important manuscript for the breast cancer community. However, given its complex statistical design( at least for some of us in the clinics with not much statistics background), simplifying the sentences might help the reader. Some of the sentences are too long.
The discussion part was good and self-explanatory but I did not see any comments about the limitations of the study.
Author Response
Reviewer 2
5 (Q) This is a good concept and a very important manuscript for the breast cancer community. However, given its complex statistical design (at least for some of us in the clinics with not much statistics background), simplifying the sentences might help the reader. Some of the sentences are too long.
(A) The manuscript has been revised critically by professional people in this field to
improve the quality of academic English writing. The long sentence has been shortened and simplified throughout the text.
6 (Q) The discussion part was good and self-explanatory but I did not see any comments about the limitations of the study.
(A) Two limitations of our study have been included in revised manuscript as follows.
“There are two limitations of the current study. Although the application of ZIP enables us to estimate the attributable proportions of three types of breast cancer survivors, personalized prediction for three types cannot be achieved without more updated information on molecular and imaging biomarkers can be included in the zero part and non-zero part, respectively. The second is related to the validation of this zero-inflated model by the application of the proposed model together with the estimated parameters to independent prospective follow-up data of this cohort in the future and also to data outside this country.” (Discussion, Page 8, Line 291 to Page 9, Line 297)

Reviewer 3 Report
Can you clarify your statement ligne 59-61 in the introduction.
Very discret spelling misstake ligne 60 overdiagnosis/ long term
very interesting article
very well written
Author Response
Reviewer 3
7 (Q) Can you clarify your statement line 59-61 in the introduction.
(A) This sentence has been revised as follows:
“The previous studies on the extent of overdiagnosis were estimated by excess incidence due to screening compared with background incidence derived from randomized control trial or predicted incidence extrapolated from previous unexposed epoch, making allowance for lead-time. Note that these previous methods, while estimating the proportion of overdiagnosis, require individual normal and incident breast cancer data and also the strong assumption of lead-time distribution. Theses traditional approaches cannot be used for assessing the impact of overdiagnosis on long-term survival while information is only available with breast cancer cases and death from breast cancer.” (Introduction, Page 2, Line 49-56).
8 (Q) Very discret spelling misstake ligne 60 overdiagnosis/ long term
(A) These misspelling errors have been corrected.
overdiagnosis (Page 2, Line 54), long-term (Page 2, Line 55)

Round 2
Reviewer 1 Report
None